# On Large-Batch Training for Deep Learning: Generalization Gap and Sharp Minima

**Nitish Shirish Keskar**[*]
Northwestern University
Evanston, IL 60208
keskar.nitish@u.northwestern.edu

**Dheevatsa Mudigere**
Intel Corporation
Bangalore, India
dheevatsa.mudigere@intel.com

**Jorge Nocedal**
Northwestern University
Evanston, IL 60208
j-nocedal@northwestern.edu

**Mikhail Smelyanskiy**
Intel Corporation
Santa Clara, CA 95054
mikhail.smelyanskiy@intel.com

**Ping Tak Peter Tang**
Intel Corporation
Santa Clara, CA 95054
peter.tang@intel.com

## Abstract

The stochastic gradient descent (SGD) method and its variants are algorithms of choice for many Deep Learning tasks. These methods operate in a small-batch regime wherein a fraction of the training data, say 32–512 data points, is sampled to compute an approximation to the gradient. It has been observed in practice that when using a larger batch there is a degradation in the quality of the model, as measured by its ability to generalize. We investigate the cause for this generalization drop in the large-batch regime and present numerical evidence that supports the view that large-batch methods tend to converge to sharp minimizers of the training and testing functions—and as is well known, sharp minima lead to poorer generalization. In contrast, small-batch methods consistently converge to flat minimizers, and our experiments support a commonly held view that this is due to the inherent noise in the gradient estimation. We discuss several strategies to attempt to help large-batch methods eliminate this generalization gap.

## 1 Introduction

Deep Learning has emerged as one of the cornerstones of large-scale machine learning. Deep Learning models are used for achieving state-of-the-art results on a wide variety of tasks including computer vision, natural language processing and reinforcement learning; see (Bengio et al., 2016) and the references therein. The problem of training these networks is one of non-convex optimization. Mathematically, this can be represented as:

$$\min_{x \in \mathbb{R}^n} \quad f(x) := \frac{1}{M} \sum_{i=1}^{M} f_i(x), \tag{1}$$

where $f_i$ is a loss function for data point $i \in \{1, 2, \cdots, M\}$ which captures the deviation of the model prediction from the data, and $x$ is the vector of weights being optimized. The process of optimizing this function is also called *training* of the network. Stochastic Gradient Descent (SGD) (Bottou, 1998; Sutskever et al., 2013) and its variants are often used for training deep networks.

---

[*]Work was performed when author was an intern at Intel Corporation

These methods minimize the objective function $f$ by iteratively taking steps of the form:

$$x_{k+1} = x_k - \alpha_k \left( \frac{1}{|B_k|} \sum_{i \in B_k} \nabla f_i(x_k) \right), \qquad (2)$$

where $B_k \subset \{1, 2, \cdots, M\}$ is the batch sampled from the data set and $\alpha_k$ is the step size at iteration $k$. These methods can be interpreted as gradient descent using noisy gradients, which and are often referred to as mini-batch gradients with batch size $|B_k|$. SGD and its variants are employed in a small-batch regime, where $|B_k| \ll M$ and typically $|B_k| \in \{32, 64, \cdots, 512\}$. These configurations have been successfully used in practice for a large number of applications; see e.g. (Simonyan & Zisserman, 2014; Graves et al., 2013; Mnih et al., 2013). Many theoretical properties of these methods are known. These include guarantees of: (a) convergence to minimizers of strongly-convex functions and to stationary points for non-convex functions (Bottou et al., 2016), (b) saddle-point avoidance (Ge et al., 2015; Lee et al., 2016), and (c) robustness to input data (Hardt et al., 2015).

Stochastic gradient methods have, however, a major drawback: owing to the sequential nature of the iteration and small batch sizes, there is limited avenue for parallelization. While some efforts have been made to parallelize SGD for Deep Learning (Dean et al., 2012; Das et al., 2016; Zhang et al., 2015), the speed-ups and scalability obtained are often limited by the small batch sizes. One natural avenue for improving parallelism is to increase the batch size $|B_k|$. This increases the amount of computation per iteration, which can be effectively distributed. However, practitioners have observed that this leads to a loss in generalization performance; see e.g. (LeCun et al., 2012). In other words, the performance of the model on testing data sets is often worse when trained with large-batch methods as compared to small-batch methods. In our experiments, we have found the drop in generalization (also called generalization gap) to be as high as $5\%$ even for smaller networks.

In this paper, we present numerical results that shed light into this drawback of large-batch methods. We observe that the generalization gap is correlated with a marked sharpness of the minimizers obtained by large-batch methods. This motivates efforts at remedying the generalization problem, as a training algorithm that employs large batches without sacrificing generalization performance would have the ability to scale to a much larger number of nodes than is possible today. This could potentially reduce the training time by orders-of-magnitude; we present an idealized performance model in the Appendix C to support this claim.

The paper is organized as follows. In the remainder of this section, we define the notation used in this paper, and in Section 2 we present our main findings and their supporting numerical evidence. In Section 3 we explore the performance of small-batch methods, and in Section 4 we briefly discuss the relationship between our results and recent theoretical work. We conclude with open questions concerning the generalization gap, sharp minima, and possible modifications to make large-batch training viable. In Appendix E, we present some attempts to overcome the problems of large-batch training.

## 1.1 NOTATION

We use the notation $f_i$ to denote the composition of loss function and a prediction function corresponding to the $i^{th}$ data point. The vector of weights is denoted by $x$ and is subscripted by $k$ to denote an iteration. We use the term small-batch (SB) method to denote SGD, or one of its variants like ADAM (Kingma & Ba, 2015) and ADAGRAD (Duchi et al., 2011), with the proviso that the gradient approximation is based on a small mini-batch. In our setup, the batch $B_k$ is randomly sampled and its size is kept fixed for every iteration. We use the term large-batch (LB) method to denote any training algorithm that uses a large mini-batch. In our experiments, ADAM is used to explore the behavior of both a small or a large batch method.

## 2 DRAWBACKS OF LARGE-BATCH METHODS

### 2.1 OUR MAIN OBSERVATION

As mentioned in Section 1, practitioners have observed a generalization gap when using large-batch methods for training deep learning models. Interestingly, this is despite the fact that large-batch methods usually yield a similar value of the training function as small-batch methods. One may put

forth the following as possible causes for this phenomenon: (i) LB methods over-fit the model; (ii) LB methods are attracted to saddle points; (iii) LB methods lack the *explorative* properties of SB methods and tend to zoom-in on the minimizer closest to the initial point; (iv) SB and LB methods converge to qualitatively different minimizers with differing generalization properties. The data presented in this paper supports the last two conjectures.

The main observation of this paper is as follows:

> The lack of generalization ability is due to the fact that large-batch methods tend to converge to *sharp minimizers* of the training function. These minimizers are characterized by a significant number of large positive eigenvalues in $\nabla^2 f(x)$, and tend to generalize less well. In contrast, small-batch methods converge to *flat minimizers* characterized by having numerous small eigenvalues of $\nabla^2 f(x)$. We have observed that the loss function landscape of deep neural networks is such that large-batch methods are attracted to regions with sharp minimizers and that, unlike small-batch methods, are unable to escape basins of attraction of these minimizers.

The concept of sharp and flat minimizers have been discussed in the statistics and machine learning literature. (Hochreiter & Schmidhuber, 1997) (informally) define a flat minimizer $\bar{x}$ as one for which the function varies slowly in a relatively large neighborhood of $\bar{x}$. In contrast, a sharp minimizer $\hat{x}$ is such that the function increases rapidly in a small neighborhood of $\hat{x}$. A flat minimum can be described with low precision, whereas a sharp minimum requires high precision. The large sensitivity of the training function at a sharp minimizer negatively impacts the ability of the trained model to generalize on new data; see Figure 1 for a hypothetical illustration. This can be explained through the lens of the minimum description length (MDL) theory, which states that statistical models that require fewer bits to describe (i.e., are of low complexity) generalize better (Rissanen, 1983). Since flat minimizers can be specified with lower precision than to sharp minimizers, they tend to have better generalization performance. Alternative explanations are proffered through the Bayesian view of learning (MacKay, 1992), and through the lens of free Gibbs energy; see e.g. Chaudhari et al. (2016).

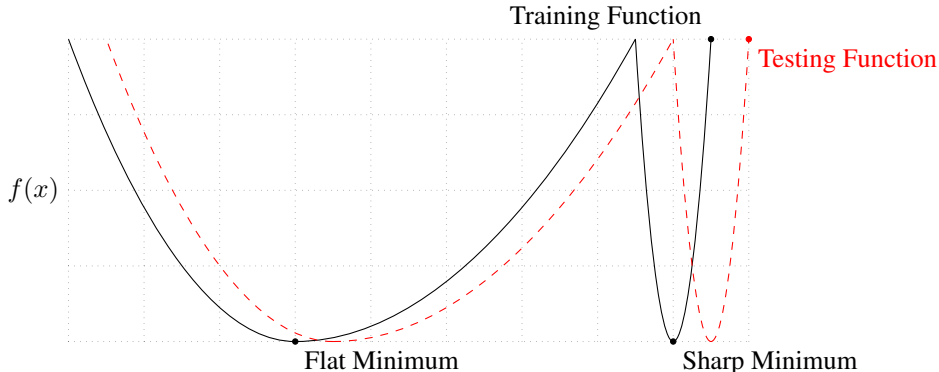

Figure 1: A Conceptual Sketch of Flat and Sharp Minima. The Y-axis indicates value of the loss function and the X-axis the variables (parameters)

## 2.2 NUMERICAL EXPERIMENTS

In this section, we present numerical results to support the observations made above. To this end, we make use of the visualization technique employed by (Goodfellow et al., 2014b) and a proposed heuristic metric of sharpness (Equation (4)). We consider 6 multi-class classification network configurations for our experiments; they are described in Table 1. The details about the data sets and network configurations are presented in Appendices A and B respectively. As is common for such problems, we use the mean cross entropy loss as the objective function $f$.

The networks were chosen to exemplify popular configurations used in practice like AlexNet (Krizhevsky et al., 2012) and VGGNet (Simonyan & Zisserman, 2014). Results on other networks

Table 1: Network Configurations

| Name | Network Type | Architecture | Data set |
|------|-------------|--------------|----------|
| $F_1$ | Fully Connected | Section B.1 | MNIST (LeCun et al., 1998a) |
| $F_2$ | Fully Connected | Section B.2 | TIMIT (Garofolo et al., 1993) |
| $C_1$ | (Shallow) Convolutional | Section B.3 | CIFAR-10 (Krizhevsky & Hinton, 2009) |
| $C_2$ | (Deep) Convolutional | Section B.4 | CIFAR-10 |
| $C_3$ | (Shallow) Convolutional | Section B.3 | CIFAR-100 (Krizhevsky & Hinton, 2009) |
| $C_4$ | (Deep) Convolutional | Section B.4 | CIFAR-100 |

and using other initialization strategies, activation functions, and data sets showed similar behavior. Since the goal of our work is not to achieve state-of-the-art accuracy or time-to-solution on these tasks but rather to characterize the *nature* of the minima for LB and SB methods, we only describe the final testing accuracy in the main paper and ignore convergence trends.

For all experiments, we used $10\%$ of the training data as batch size for the large-batch experiments and 256 data points for small-batch experiments. We used the ADAM optimizer for both regimes. Experiments with other optimizers for the large-batch experiments, including ADAGRAD (Duchi et al., 2011), SGD (Sutskever et al., 2013) and adaQN (Keskar & Berahas, 2016), led to similar results. All experiments were conducted 5 times from different (uniformly distributed random) starting points and we report both mean and standard-deviation of measured quantities. The baseline performance for our setup is presented Table 2. From this, we can observe that on all networks, both approaches led to high training accuracy but there is a significant difference in the generalization performance. The networks were trained, without any budget or limits, until the loss function ceased to improve.

Table 2: Performance of small-batch (SB) and large-batch (LB) variants of ADAM on the 6 networks listed in Table 1

| Name | Training Accuracy | | Testing Accuracy | |
|------|------|------|------|------|
| | SB | LB | SB | LB |
| $F_1$ | $99.66\% \pm 0.05\%$ | $99.92\% \pm 0.01\%$ | $98.03\% \pm 0.07\%$ | $97.81\% \pm 0.07\%$ |
| $F_2$ | $99.99\% \pm 0.03\%$ | $98.35\% \pm 2.08\%$ | $64.02\% \pm 0.2\%$ | $59.45\% \pm 1.05\%$ |
| $C_1$ | $99.89\% \pm 0.02\%$ | $99.66\% \pm 0.2\%$ | $80.04\% \pm 0.12\%$ | $77.26\% \pm 0.42\%$ |
| $C_2$ | $99.99\% \pm 0.04\%$ | $99.99\% \pm 0.01\%$ | $89.24\% \pm 0.12\%$ | $87.26\% \pm 0.07\%$ |
| $C_3$ | $99.56\% \pm 0.44\%$ | $99.88\% \pm 0.30\%$ | $49.58\% \pm 0.39\%$ | $46.45\% \pm 0.43\%$ |
| $C_4$ | $99.10\% \pm 1.23\%$ | $99.57\% \pm 1.84\%$ | $63.08\% \pm 0.5\%$ | $57.81\% \pm 0.17\%$ |

We emphasize that the generalization gap is not due to *over-fitting* or *over-training* as commonly observed in statistics. This phenomenon manifest themselves in the form of a testing accuracy curve that, at a certain iterate peaks, and then decays due to the model learning idiosyncrasies of the training data. This is not what we observe in our experiments; see Figure 2 for the training–testing curve of the $F_2$ and $C_1$ networks, which are representative of the rest. As such, early-stopping heuristics aimed at preventing models from over-fitting would not help reduce the generalization gap. The difference between the *training and testing* accuracies for the networks is due to the specific choice of the network (e.g. AlexNet, VGGNet etc.) and is not the focus of this study. Rather, our goal is to study the source of the testing performance disparity of the two regimes, SB and LB, on a given network model.

### 2.2.1 PARAMETRIC PLOTS

We first present parametric 1-D plots of the function as described in (Goodfellow et al., 2014b). Let $x_s^\star$ and $x_\ell^\star$ indicate the solutions obtained by running ADAM using small and large batch sizes respectively. We plot the loss function, on both training and testing data sets, along a line-segment containing the two points. Specifically, for $\alpha \in [-1, 2]$, we plot the function $f(\alpha x_\ell^\star + (1 - \alpha)x_s^\star)$ and also superimpose the classification accuracy at the intermediate points; see Figure 3[1]. For this

---

[1]The code to reproduce the parametric plot on exemplary networks can be found in our GitHub repository: `https://github.com/keskarnitish/large-batch-training`.

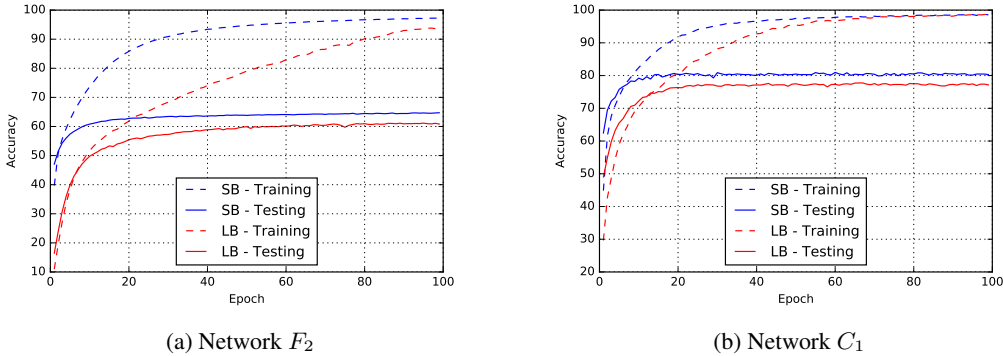

(a) Network $F_2$ (b) Network $C_1$

Figure 2: Training and testing accuracy for SB and LB methods as a function of epochs.

experiment, we randomly chose a pair of SB and LB minimizers from the 5 trials used to generate the data in Table 2. The plots show that the LB minima are strikingly sharper than the SB minima in this one-dimensional manifold. The plots in Figure 3 only explore a linear slice of the function, but in Figure 7 in Appendix D, we plot $f(\sin(\frac{\alpha\pi}{2})x_\ell^\star + \cos(\frac{\alpha\pi}{2})x_s^\star)$ to monitor the function along a curved path between the two minimizers . There too, the relative sharpness of the minima is evident.

### 2.2.2 Sharpness of Minima

So far, we have used the term *sharp minimizer* loosely, but we noted that this concept has received attention in the literature (Hochreiter & Schmidhuber, 1997). Sharpness of a minimizer can be characterized by the magnitude of the eigenvalues of $\nabla^2 f(x)$, but given the prohibitive cost of this computation in deep learning applications, we employ a sensitivity measure that, although imperfect, is computationally feasible, even for large networks. It is based on exploring a small neighborhood of a solution and computing the largest value that the function $f$ can attain in that neighborhood. We use that value to measure the sensitivity of the training function at the given local minimizer. Now, since the maximization process is not accurate, and to avoid being mislead by the case when a large value of $f$ is attained only in a tiny subspace of $\mathbb{R}^n$, we perform the maximization both in the entire space $\mathbb{R}^n$ as well as in random manifolds. For that purpose, we introduce an $n \times p$ matrix $A$, whose columns are randomly generated. Here $p$ determines the dimension of the manifold, which in our experiments is chosen as $p = 100$.

Specifically, let $\mathcal{C}_\epsilon$ denote a box around the solution over which the maximization of $f$ is performed, and let $A \in \mathbb{R}^{n \times p}$ be the matrix defined above. In order to ensure invariance of sharpness to problem dimension and sparsity, we define the constraint set $\mathcal{C}_\epsilon$ as:

$$\mathcal{C}_\epsilon = \{z \in \mathbb{R}^p : -\epsilon(|(A^+x)_i| + 1) \le z_i \le \epsilon(|(A^+x)_i| + 1) \quad \forall i \in \{1, 2, \cdots, p\}\}, \qquad (3)$$

where $A^+$ denotes the pseudo-inverse of $A$. Thus $\epsilon$ controls the size of the box. We can now define our measure of sharpness (or sensitivity).

**Metric 2.1.** *Given $x \in \mathbb{R}^n$, $\epsilon > 0$ and $A \in \mathbb{R}^{n \times p}$, we define the $(\mathcal{C}_\epsilon, A)$-sharpness of $f$ at $x$ as:*

$$\phi_{x,f}(\epsilon, A) := \frac{(\max_{y \in \mathcal{C}_\epsilon} f(x + Ay)) - f(x)}{1 + f(x)} \times 100. \qquad (4)$$

Unless specified otherwise, we use this metric for sharpness for the rest of the paper; if $A$ is not specified, it is assumed to be the identity matrix, $I_n$. (We note in passing that, in the convex optimization literature, the term sharp minimum has a different definition (Ferris, 1988), but that concept is not useful for our purposes.)

In Tables 3 and 4, we present the values of the sharpness metric (4) for the minimizers of the various problems. Table 3 explores the full-space (i.e., $A = I_n$) whereas Table 4 uses a randomly sampled $n \times 100$ dimensional matrix $A$. We report results with two values of $\epsilon$, $(10^{-3}, 5 \cdot 10^{-4})$. In all experiments, we solve the maximization problem in Equation (4) inexactly by applying 10 iterations of L-BFGS-B (Byrd et al., 1995). This limit on the number of iterations was necessitated by the

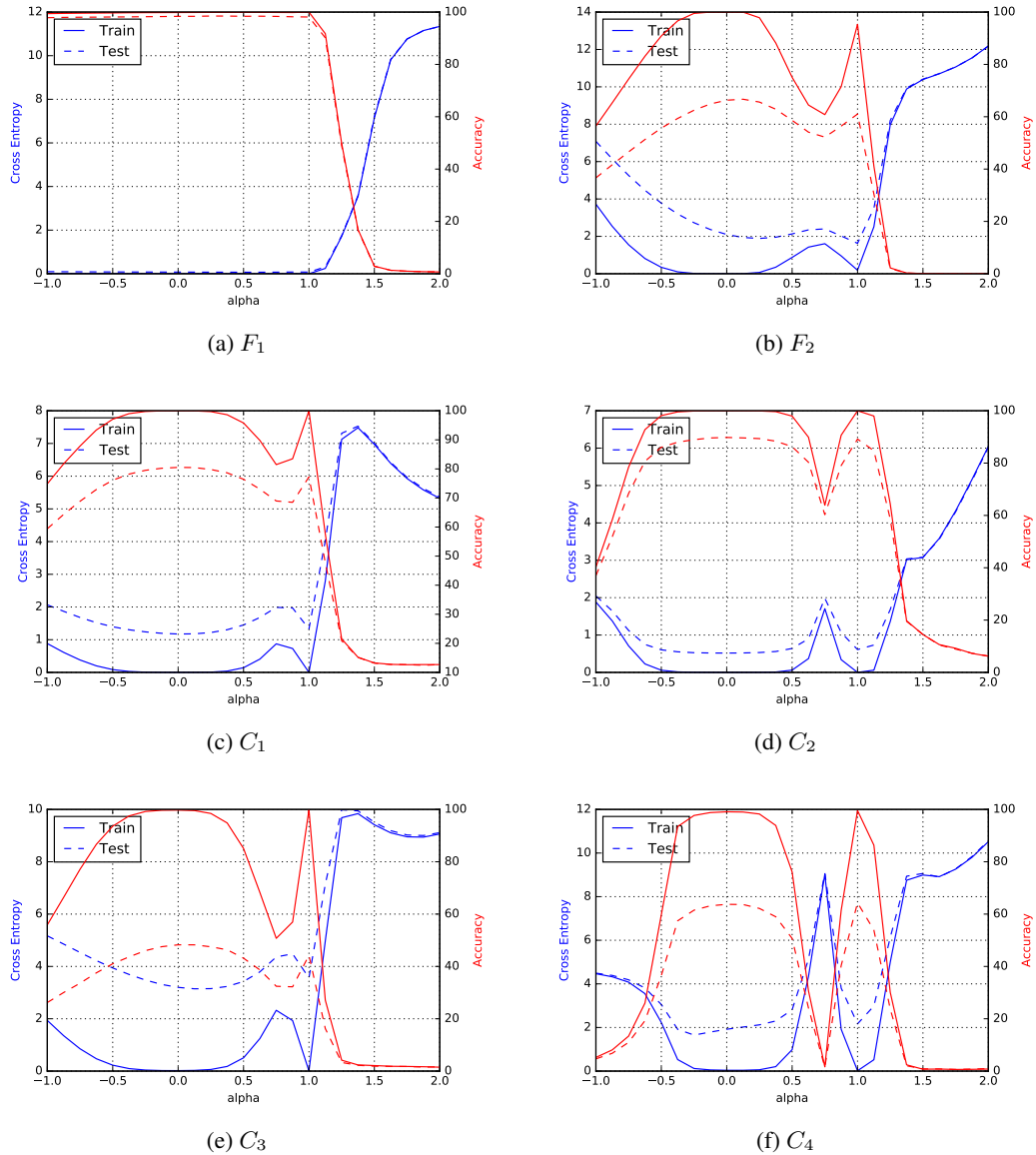

(a) $F_1$    (b) $F_2$

(c) $C_1$    (d) $C_2$

(e) $C_3$    (f) $C_4$

Figure 3: Parametric Plots – Linear (Left vertical axis corresponds to cross-entropy loss, $f$, and right vertical axis corresponds to classification accuracy; solid line indicates training data set and dashed line indicated testing data set); $\alpha = 0$ corresponds to the SB minimizer and $\alpha = 1$ to the LB minimizer.

large cost of evaluating the true objective $f$. Both tables show a 1–2 order-of-magnitude difference between the values of our metric for the SB and LB regimes. These results reinforce the view that the solutions obtained by a large-batch method defines points of larger sensitivity of the training function. In Appedix E, we describe approaches to attempt to remedy this generalization problem of LB methods. These approaches include data augmentation, conservative training and adversarial training. Our preliminary findings show that these approaches help reduce the generalization gap but still lead to relatively sharp minimizers and as such, do not completely remedy the problem.

Note that Metric 2.1 is closely related to the spectrum of $\nabla^2 f(x)$. Assuming $\epsilon$ to be small enough, when $A = I_n$, the value (4) relates to the largest eigenvalue of $\nabla^2 f(x)$ and when $A$ is randomly sampled it approximates the Ritz value of $\nabla^2 f(x)$ projected onto the column-space of $A$.

Table 3: Sharpness of Minima in Full Space; $\epsilon$ is defined in (3).

|       | $\epsilon = 10^{-3}$ | | $\epsilon = 5 \cdot 10^{-4}$ | |
|-------|------------------|-------------------|------------------|--------------------|
|       | SB               | LB                | SB               | LB                 |
| $F_1$ | $1.23 \pm 0.83$  | $205.14 \pm 69.52$ | $0.61 \pm 0.27$ | $42.90 \pm 17.14$  |
| $F_2$ | $1.39 \pm 0.02$  | $310.64 \pm 38.46$ | $0.90 \pm 0.05$ | $93.15 \pm 6.81$   |
| $C_1$ | $28.58 \pm 3.13$ | $707.23 \pm 43.04$ | $7.08 \pm 0.88$ | $227.31 \pm 23.23$ |
| $C_2$ | $8.68 \pm 1.32$  | $925.32 \pm 38.29$ | $2.07 \pm 0.86$ | $175.31 \pm 18.28$ |
| $C_3$ | $29.85 \pm 5.98$ | $258.75 \pm 8.96$  | $8.56 \pm 0.99$ | $105.11 \pm 13.22$ |
| $C_4$ | $12.83 \pm 3.84$ | $421.84 \pm 36.97$ | $4.07 \pm 0.87$ | $109.35 \pm 16.57$ |

Table 4: Sharpness of Minima in Random Subspaces of Dimension 100

|       | $\epsilon = 10^{-3}$ | | $\epsilon = 5 \cdot 10^{-4}$ | |
|-------|------------------|--------------------|------------------|-------------------|
|       | SB               | LB                 | SB               | LB                |
| $F_1$ | $0.11 \pm 0.00$  | $9.22 \pm 0.56$    | $0.05 \pm 0.00$  | $9.17 \pm 0.14$   |
| $F_2$ | $0.29 \pm 0.02$  | $23.63 \pm 0.54$   | $0.05 \pm 0.00$  | $6.28 \pm 0.19$   |
| $C_1$ | $2.18 \pm 0.23$  | $137.25 \pm 21.60$ | $0.71 \pm 0.15$  | $29.50 \pm 7.48$  |
| $C_2$ | $0.95 \pm 0.34$  | $25.09 \pm 2.61$   | $0.31 \pm 0.08$  | $5.82 \pm 0.52$   |
| $C_3$ | $17.02 \pm 2.20$ | $236.03 \pm 31.26$ | $4.03 \pm 1.45$  | $86.96 \pm 27.39$ |
| $C_4$ | $6.05 \pm 1.13$  | $72.99 \pm 10.96$  | $1.89 \pm 0.33$  | $19.85 \pm 4.12$  |

We conclude this section by noting that the sharp minimizers identified in our experiments do not resemble a cone, i.e., the function does not increase rapidly along all (or even most) directions. By sampling the loss function in a neighborhood of LB solutions, we observe that it rises steeply only along a small dimensional subspace (e.g. 5% of the whole space); on most other directions, the function is relatively flat.

## 3 SUCCESS OF SMALL-BATCH METHODS

It is often reported that when increasing the batch size for a problem, there exists a threshold after which there is a deterioration in the quality of the model. This behavior can be observed for the $F_2$ and $C_1$ networks in Figure 4. In both of these experiments, there is a batch size ($\approx 15000$ for $F_2$ and $\approx 500$ for $C_1$) after which there is a large drop in testing accuracy. Notice also that the upward drift in value of the sharpness is considerably reduced around this threshold. Similar thresholds exist for the other networks in Table 1.

Let us now consider the behavior of SB methods, which use noisy gradients in the step computation. From the results reported in the previous section, it appears that noise in the gradient pushes the iterates out of the basin of attraction of sharp minimizers and encourages movement towards a flatter minimizer where noise will not cause exit from that basin. When the batch size is greater than the threshold mentioned above, the noise in the stochastic gradient is not sufficient to cause ejection from the initial basin leading to convergence to sharper a minimizer.

To explore that in more detail, consider the following experiment. We train the network for 100 epochs using ADAM with a batch size of 256, and retain the iterate after each epoch in memory. Using these 100 iterates as starting points we train the network using a LB method for 100 epochs and receive a 100 *piggybacked* (or warm-started) large-batch solutions. We plot in Figure 5 the testing accuracy and sharpness of these large-batch solutions, along with the testing accuracy of the small-batch iterates. Note that when warm-started with only a few initial epochs, the LB method does not yield a generalization improvement. The concomitant sharpness of the iterates also stays high. On the other hand, after certain number of epochs of warm-starting, the accuracy improves and sharpness of the large-batch iterates drop. This happens, apparently, when the SB method has ended its exploration phase and discovered a flat minimizer; the LB method is then able to converge towards it, leading to good testing accuracy.

It has been speculated that LB methods tend to be attracted to minimizers close to the starting point $x_0$, whereas SB methods move away and locate minimizers that are farther away. Our numerical

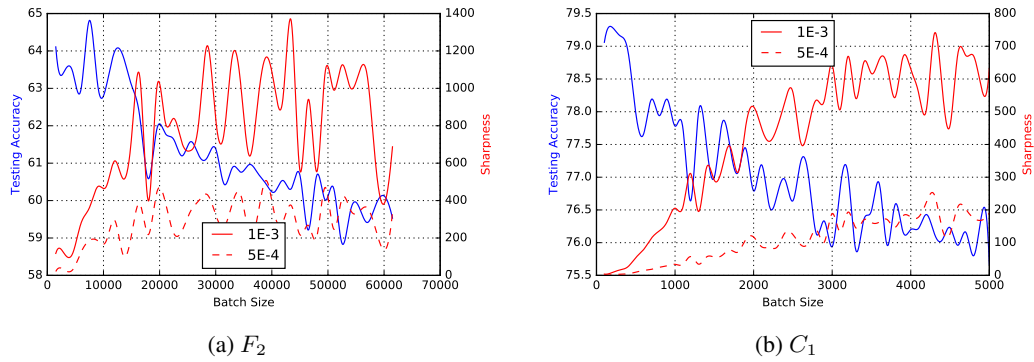

(a) $F_2$ (b) $C_1$

Figure 4: Testing Accuracy and Sharpness v/s Batch Size. The X-axis corresponds to the batch size used for training the network for $100$ epochs, left Y-axis corresponds to the testing accuracy at the final iterate and right Y-axis corresponds to the sharpness of that iterate. We report sharpness for two values of $\epsilon$: $10^{-3}$ and $5 \cdot 10^{-4}$.

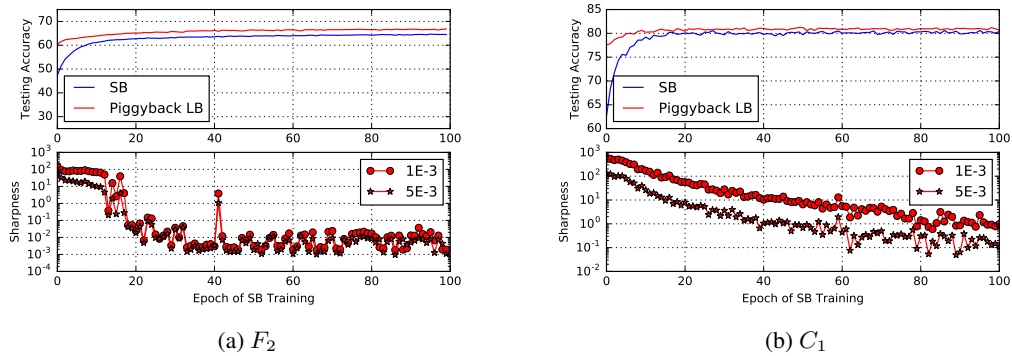

(a) $F_2$ (b) $C_1$

Figure 5: Warm-starting experiments. The upper figures report the testing accuracy of the SB method (blue line) and the testing accuracy of the warm started (piggybacked) LB method (red line), as a function of the number of epochs of the SB method. The lower figures plot the sharpness measure (4) for the solutions obtained by the piggybacked LB method v/s the number of warm-starting epochs of the SB method.

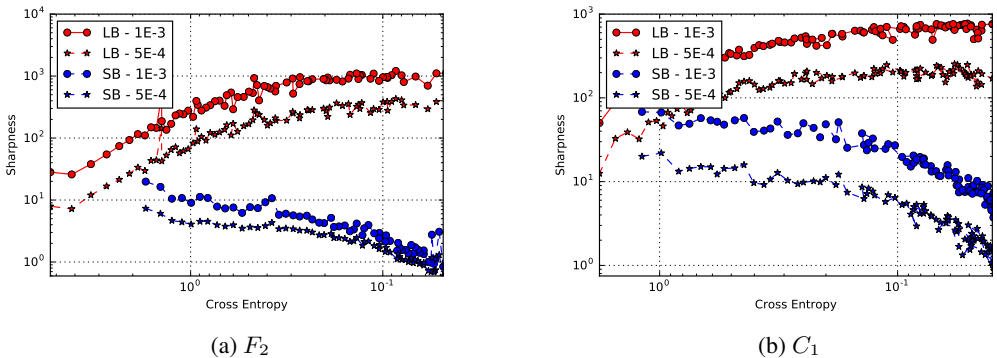

(a) $F_2$                                        (b) $C_1$

Figure 6: Sharpness v/s Cross Entropy Loss for SB and LB methods.

experiments support this view: we observed that the ratio of $\|x_s^\star - x_0\|_2$ and $\|x_\ell^\star - x_0\|_2$ was in the range of 3–10.

In order to further illustrate the qualitative difference between the solutions obtained by SB and LB methods, we plot in Figure 6 our sharpness measure (4) against the loss function (cross entropy) for one random trial of the $F_2$ and $C_1$ networks. For larger values of the loss function, i.e., near the initial point, SB and LB method yield similar values of sharpness. As the loss function reduces, the sharpness of the iterates corresponding to the LB method rapidly increases, whereas for the SB method the sharpness stays relatively constant initially and then reduces, suggesting an exploration phase followed by convergence to a flat minimizer.

## 4    DISCUSSION AND CONCLUSION

In this paper, we present numerical experiments that support the view that convergence to sharp minimizers gives rise to the poor generalization of large-batch methods for deep learning. To this end, we provide one-dimensional parametric plots and perturbation (sharpness) measures for a variety of deep learning architectures. In Appendix E, we describe our attempts to remedy the problem, including data augmentation, conservative training and robust optimization. Our preliminary investigation suggests that these strategies do not correct the problem; they improve the generalization of large-batch methods but still lead to relatively sharp minima. Another prospective remedy includes the use of *dynamic sampling* where the batch size is increased gradually as the iteration progresses (Byrd et al., 2012; Friedlander & Schmidt, 2012). The potential viability of this approach is suggested by our warm-starting experiments (see Figure 5) wherein high testing accuracy is achieved using a large-batch method that is warm-start with a small-batch method.

Recently, a number of researchers have described interesting theoretical properties of the loss surface of deep neural networks; see e.g. (Choromanska et al., 2015; Soudry & Carmon, 2016; Lee et al., 2016). Their work shows that, under certain regularity assumptions, the loss function of deep learning models is fraught with many local minimizers and that many of these minimizers correspond to a similar loss function value. Our results are in alignment these observations since, in our experiments, both sharp and flat minimizers have very similar loss function values. We do not know, however, if the theoretical models mentioned above provide information about the existence and density of sharp minimizers of the loss surface.

Our results suggest some questions: (a) can one *prove* that large-batch (LB) methods typically converge to sharp minimizers of deep learning training functions? (In this paper, we only provided some numerical evidence.); (b) what is the relative density of the two kinds of minima?; (c) can one design neural network architectures for various tasks that are suitable to the properties of LB methods?; (d) can the networks be initialized in a way that enables LB methods to succeed?; (e) is it possible, through algorithmic or regulatory means to steer LB methods away from sharp minimizers?

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

# A    DETAILS ABOUT DATA SETS

We summarize the data sets used in our experiments in Table 5. TIMIT is a speech recognition data set which is pre-processed using Kaldi (Povey et al., 2011) and trained using a fully-connected network. The rest of the data sets are used without any pre-processing.

Table 5: Data Sets

| Data Set | # Data Points | | # Features | # Classes | Reference |
|---|---|---|---|---|---|
| | Train | Test | | | |
| MNIST | 60000 | 10000 | $28 \times 28$ | 10 | (LeCun et al., 1998a;b) |
| TIMIT | 721329 | 310621 | 360 | 1973 | (Garofolo et al., 1993) |
| CIFAR-10 | 50000 | 10000 | $32 \times 32$ | 10 | (Krizhevsky & Hinton, 2009) |
| CIFAR-100 | 50000 | 10000 | $32 \times 32$ | 100 | (Krizhevsky & Hinton, 2009) |

# B    ARCHITECTURE OF NETWORKS

## B.1    NETWORK $F_1$

For this network, we use a 784-dimensional input layer followed by 5 batch-normalized (Ioffe & Szegedy, 2015) layers of 512 neurons each with ReLU activations. The output layer consists of 10 neurons with the softmax activation.

## B.2    NETWORK $F_2$

The network architecture for $F_2$ is similar to $F_1$. We use a 360-dimensional input layer followed by 7 batch-normalized layers of 512 neurons with ReLU activation. The output layer consists of 1973 neurons with the softmax activation.

## B.3    NETWORKS $C_1$ AND $C_3$

The $C_1$ network is a modified version of the popular AlexNet configuration (Krizhevsky et al., 2012). For simplicity, denote a stack of $n$ convolution layers of $a$ filters and a Kernel size of $b \times c$ with stride length of $d$ as $n \times [a, b, c, d]$. The $C_1$ configuration uses 2 sets of $[64, 5, 5, 2]$–MaxPool(3) followed by 2 dense layers of sizes $(384, 192)$ and finally, an output layer of size 10. We use batch-normalization for all layers and ReLU activations. We also use Dropout (Srivastava et al., 2014) of 0.5 retention probability for the two dense layers. The configuration $C_3$ is identical to $C_1$ except it uses 100 softmax outputs instead of 10.

## B.4    NETWORKS $C_2$ AND $C_4$

The $C_2$ network is a modified version of the popular VGG configuration (Simonyan & Zisserman, 2014). The $C_3$ network uses the configuration: $2 \times [64, 3, 3, 1], 2 \times [128, 3, 3, 1], 3 \times [256, 3, 3, 1], 3 \times [512, 3, 3, 1], 3 \times [512, 3, 3, 1]$ which a MaxPool(2) after each stack. This stack is followed by a 512-dimensional dense layer and finally, a 10-dimensional output layer. The activation and properties of each layer is as in B.3. As is the case with $C_3$ and $C_1$, the configuration $C_4$ is identical to $C_2$ except that it uses 100 softmax outputs instead of 10.

## C    PERFORMANCE MODEL

As mentioned in Section 1, a training algorithm that operates in the large-batch regime without suffering from a generalization gap would have the ability to scale to much larger number of nodes than is currently possible. Such and algorithm might also improve training time through faster convergence. We present an idealized performance model that demonstrates our goal.

For LB method to be competitive with SB method, the LB method must (i) converge to minimizers that generalize well, and (ii) do it in a reasonably number of iterations, which we analyze here. Let $I_s$ and $I_\ell$ be number of iterations required by SB and LB methods to reach the point of comparable test accuracy, respectively. Let $B_s$ and $B_\ell$ be corresponding batch sizes and $P$ be number of processors being used for training. Assume that $P < B_\ell$, and let $f_s(P)$ be the parallel efficiency of the SB method. For simplicity, we assume that $f_\ell(P)$, the parallel efficiency of the LB method, is 1.0. In other words, we assume that the LB method is perfectly scalable due to use of a large batch size.

For LB to be faster than SB, we must have

$$I_\ell \frac{B_\ell}{P} < I_s \frac{B_s}{P f_s(P)}.$$

In other words, the ratio of iterations of LB to the iterations of SB should be

$$\frac{I_\ell}{I_s} < \frac{B_s}{f_s(P) B_\ell}.$$

For example, if $f_s(P) = 0.2$ and $B_s/B_\ell = 0.1$, the LB method must converge in at most half as many iterations as the SB method to see performance benefits. We refer the reader to (Das et al., 2016) for a more detailed model and a commentary on the effect of batch-size on the performance.

## D    CURVILINEAR PARAMETRIC PLOTS

The parametric plots for the curvilinear path from $x_s^\star$ to $x_\ell^\star$, i.e., $f(\sin(\frac{\alpha\pi}{2})x_\ell^\star + \cos(\frac{\alpha\pi}{2})x_s^\star)$ can be found in Figure 7.

## E    ATTEMPTS TO IMPROVE LB METHODS

In this section, we discuss a few strategies that aim to remedy the problem of poor generalization for large-batch methods. As in Section 2, we use $10\%$ as the percentage batch-size for large-batch experiments and 256 for small-batch methods. For all experiments, we use ADAM as the optimizer irrespective of batch-size.

### E.1    DATA AUGMENTATION

Given that large-batch methods appear to be attracted to sharp minimizers, one can ask whether it is possible to modify the geometry of the loss function so that it is more benign to large-batch methods. The loss function depends both on the geometry of the objective function and to the size and properties of the training set. One approach we consider is data augmentation; see e.g. (Krizhevsky et al., 2012; Simonyan & Zisserman, 2014). The application of this technique is domain specific but generally involves augmenting the data set through controlled modifications on the training data. For instance, in the case of image recognition, the training set can be augmented through translations, rotations, shearing and flipping of the training data. This technique leads to regularization of the network and has been employed for improving testing accuracy on several data sets.

In our experiments, we train the 4 image-based (convolutional) networks using aggressive data augmentation and present the results in Table 6. For the augmentation, we use horizontal reflections, random rotations up to $10°$ and random translation of up to $0.2$ times the size of the image. It is evident from the table that, while the LB method achieves accuracy comparable to the SB method (also with training data augmented), the sharpness of the minima still exists, suggesting sensitivity to images contained in neither training or testing set. In this section, we exclude parametric plots and sharpness values for the SB method owing to space constraints and the similarity to those presented in Section 2.2.

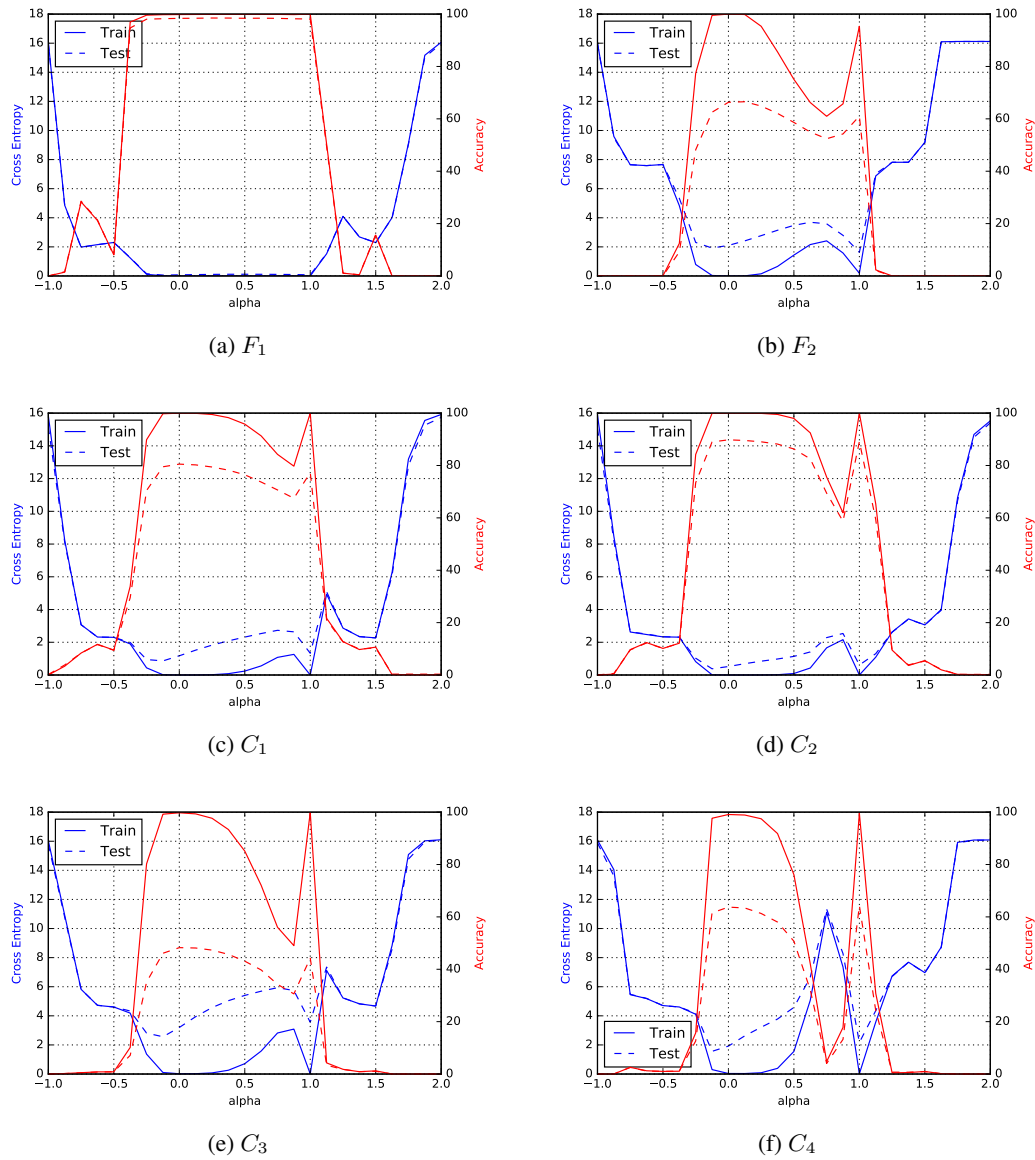

Figure 7: Parametric Plots – Curvilinear (Left vertical axis corresponds to cross-entropy loss, $f$, and right vertical axis corresponds to classification accuracy; solid line indicates training data set and dashed line indicated testing data set); $\alpha = 0$ corresponds to the SB minimizer while $\alpha = 1$ corresponds to the LB minimizer

Table 6: Effect of Data Augmentation

|  | Testing Accuracy | | Sharpness (LB method) | |
|  | Baseline (SB) | Augmented LB | $\epsilon = 10^{-3}$ | $\epsilon = 5 \cdot 10^{-4}$ |
|---|---|---|---|---|
| $C_1$ | $83.63\% \pm 0.14\%$ | $82.50\% \pm 0.67\%$ | $231.77 \pm 30.50$ | $45.89 \pm 3.83$ |
| $C_2$ | $89.82\% \pm 0.12\%$ | $90.26\% \pm 1.15\%$ | $468.65 \pm 47.86$ | $105.22 \pm 19.57$ |
| $C_3$ | $54.55\% \pm 0.44\%$ | $53.03\% \pm 0.33\%$ | $103.68 \pm 11.93$ | $37.67 \pm 3.46$ |
| $C_4$ | $63.05\% \pm 0.5\%$ | $65.88 \pm 0.13\%$ | $271.06 \pm 29.69$ | $45.31 \pm 5.93$ |

Table 7: Effect of Conservative Training

| | Testing Accuracy | | Sharpness (LB method) | |
|---|---|---|---|---|
| | Baseline (SB) | Conservative LB | $\epsilon = 10^{-3}$ | $\epsilon = 5 \cdot 10^{-4}$ |
| $F_1$ | $98.03\% \pm 0.07\%$ | $98.12\% \pm 0.01\%$ | $232.25 \pm 63.81$ | $46.02 \pm 12.58$ |
| $F_2$ | $64.02\% \pm 0.2\%$ | $61.94\% \pm 1.10\%$ | $928.40 \pm 51.63$ | $190.77 \pm 25.33$ |
| $C_1$ | $80.04\% \pm 0.12\%$ | $78.41\% \pm 0.22\%$ | $520.34 \pm 34.91$ | $171.19 \pm 15.13$ |
| $C_2$ | $89.24\% \pm 0.05\%$ | $88.495\% \pm 0.63\%$ | $632.01 \pm 208.01$ | $108.88 \pm 47.36$ |
| $C_3$ | $49.58\% \pm 0.39\%$ | $45.98\% \pm 0.54\%$ | $337.92 \pm 33.09$ | $110.69 \pm 3.88$ |
| $C_4$ | $63.08\% \pm 0.10\%$ | $62.51 \pm 0.67$ | $354.94 \pm 20.23$ | $68.76 \pm 16.29$ |

## E.2 CONSERVATIVE TRAINING

In (Li et al., 2014), the authors argue that the convergence rate of SGD for the large-batch setting can be improved by obtaining iterates through the following proximal sub-problem.

$$x_{k+1} = \arg\min_x \frac{1}{|B_k|} \sum_{i \in B_k} f_i(x) + \frac{\lambda}{2}\|x - x_k\|_2^2 \tag{5}$$

The motivation for this strategy is, in the context of large-batch methods, to better utilize a batch before moving onto the next one. The minimization problem is solved inexactly using 3–5 iterations of gradient descent, co-ordinate descent or L-BFGS. (Li et al., 2014) report that this not only improves the convergence rate of SGD but also leads to improved empirical performance on convex machine learning problems. The underlying idea of utilizing a batch is not specific to convex problems and we can apply the same framework for deep learning, however, without theoretical guarantees. Indeed, similar algorithms were proposed in (Zhang et al., 2015) and (Mobahi, 2016) for Deep Learning. The former placed emphasis on parallelization of small-batch SGD and asynchrony while the latter on a diffusion-continuation mechanism for training. The results using the conservative training approach are presented in Figure 7. In all experiments, we solve the problem (5) using 3 iterations of ADAM and set the regularization parameter $\lambda$ to be $10^{-3}$. Again, there is a statistically significant improvement in the testing accuracy of the large-batch method but it does not solve the problem of sensitivity.

## E.3 ROBUST TRAINING

A natural way of avoiding sharp minima is through *robust optimization* techniques. These methods attempt to optimize a worst-case cost as opposed to the nominal (or true) cost. Mathematically, given an $\epsilon > 0$, these techniques solve the problem

$$\min_x \quad \phi(x) := \max_{\|\Delta x\| \leq \epsilon} f(x + \Delta x) \tag{6}$$

Geometrically, classical (nominal) optimization attempts to locate the lowest point of a valley, while robust optimization attempts to lower an $\epsilon$–disc down the loss surface. We refer an interested reader to (Bertsimas et al., 2010), and the references therein, for a review of non-convex robust optimization. A direct application of this technique is, however, not feasible in our context since each iteration is prohibitively expensive because it involves solving a large-scale second-order conic program (SOCP).

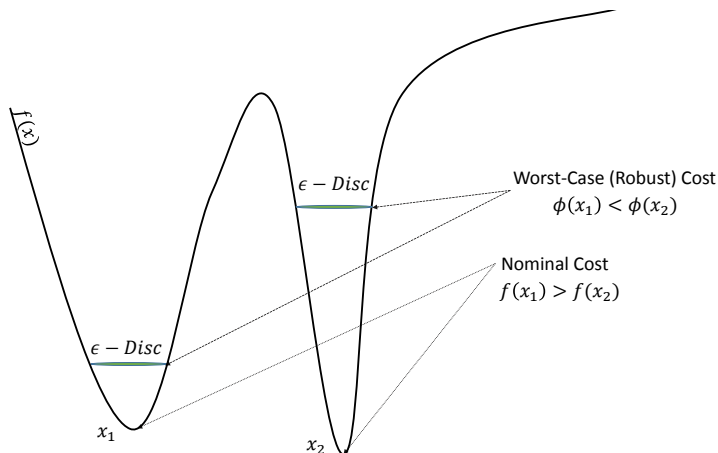

Figure 8: Illustration of Robust Optimization

In the context of Deep Learning, there are two inter-dependent forms of robustness: robustness to the data and robustness to the solution. The former exploits the fact that the function $f$ is inherently a statistical model, while the latter treats $f$ as a black-box function. In (Shaham et al., 2015), the authors prove the equivalence between robustness of the solution (with respect to the data) and adversarial training (Goodfellow et al., 2014a).

Given the partial success of the data augmentation strategy, it is natural to question the efficacy of adversarial training. As described in (Goodfellow et al., 2014a), adversarial training also aims to artificially increase the training set but, unlike randomized data augmentation, uses the model's sensitivity to construct new examples. Despite its intuitive appeal, in our experiments, we found that this strategy did not improve generalization. Similarly, we observed no generalization benefit from the stability training proposed by (Zheng et al., 2016). In both cases, the testing accuracy, sharpness values and the parametric plots were similar to the unmodified (baseline) case discussed in Section 2. It remains to be seen whether adversarial training (or any other form of robust training) can increase the viability of large-batch training.

