# Peer review of "On Large-Batch Training for Deep Learning: Generalization Gap and Sharp Minima"

_ICLR 2017 — accepted_

[Official Review · AnonReviewer2 · rating 10 · confidence 3 · 16 Dec 2016]
**Little novelty but valuable empirical evidence**
originality 4

The paper is an empirical study to justify that: 1. SGD with smaller batch sizes converges to flatter minima, 2. flatter minima have better generalization ability. 

Pros and Cons:
Although there is little novelty in the paper, I think the work is of great value in shedding light into some interesting questions around generalization of deep networks. 

Significance:
I think such results may have impact on both theory and practice, respectively by suggesting what assumptions are legitimate for real scenarios for building new theories, or be used heuristically to develop new algorithms with generalization by smart manipulation of mini-batch sizes.

Comments:
Earlier I had some concern about the correctness of a claim made by the authors, which is resolved now. They had claimed their proposed sharpness criterion is scale invariance. They took care of it by removing this claim in the revised version.

[Official Review · AnonReviewer3 · rating 6 · confidence 4 · 16 Dec 2016]
**Good paper**

I think that the paper is quite interesting and useful. 
It might benefit from additional investigations, e.g., by adding some rescaled Gaussian noise to gradients during the LB regime one can get advantages of the SB regime.

[Public Comment · Alex Lamb · 17 Dec 2016]
**Good paper**

I think that this is a great empirical exploration of long-held folk wisdom in the Deep Learning community - that using larger minibatches makes generalization error worse.  

The paper does a good of explaining why phenomenon occurs, by analyzing the "sharpness" of the loss function for large-batch and small-batch trained models.  

Some other connections that I think would be interesting to explore: 
  -If you use very small minibatches, does generalization get even better (perhaps at the expense of very slow training).  
  -Can other forms of noise injection compensate for the use of a larger minibatch?  For example, if I inject increasing amounts of noise into the gradients or the parameters with larger batches, does this close the gap with small-batch training?  
  -The motivation for using smaller minibatches here seems closely related to the motivation for adversarial examples (ensuring that loss is relatively flat in a large region around data points).

[Official Review · AnonReviewer1 · rating 8 · confidence 3 · 20 Dec 2016]
**Analysis of large batch training**
clarity 4 · impact 4

Interesting paper, definitely provides value to the community by discussing why large batch gradient descent does not work too well

[Final Decision · Program Chairs · 06 Feb 2017]
**ICLR committee final decision**

All reviews (including the public one) were extremely positive, and this sheds light on a universal engineering issue that arises in fitting non-convex models. I think the community will benefit a lot from the insights here.